# Investigations on the Mechanical Response of Gradient Lattice Structures Manufactured via SLM

**Judyta Sienkiewicz** [1,*] **, Paweł Płatek** [1] **, Fengchun Jiang** [2] **, Xiaojing Sun** [2] **and
Alexis Rusinek** [3,4]

1   Institute of Armament Technology, Military University of Technology, Kaliskiego 2, 00-908 Warsaw, Poland;
    pawel.platek@wat.edu.pl
2   College of Material Science and Chemical Engineering, Harbin Engineering University, 145 Nan-Tong Street,
    Harbin 150001, China; fengchunjiang@hrbeu.edu.cn (F.J.); sunxiaojing@hrbeu.edu.cn (X.S.)
3   Laboratory of Microstructure Studies and Mechanics of Materials, UMR-CNRS 7239, Lorraine University,
    7 rue Félix Savart, BP 15082, 57073 Metz, CEDEX 03, France; alexis.rusinek@univ-lorraine.fr
4   Chair of Excellence, Departamento de Ingeniería Mecánica, UC3M (Universidad Carlos III de Madrid) Avda.
    de la Universidad 30, 28911 Leganés, Madrid, Spain
*   Correspondence: judyta.sienkiewicz@wat.edu.pl; Tel.: +48-261-837-494

**Abstract:** The main aim of the paper is to evaluate the mechanical behavior or lattice specimens subjected to quasi-static and dynamic compression tests. Both regular and three different variants of SS 316L lattice structures with gradually changed topologies (discrete, increase and decrease) have been successfully designed and additively manufactured with the use of the selective laser melting technique. The fabricated structures were subjected to geometrical quality control, microstructure analysis, phase characterization and compression tests under quasi-static and dynamic loading conditions. The mismatch between dimensions in the designed and produced lattices was noticed. It generally results from the adopted technique of the manufacturing process. The microstructure and phase composition were in good agreement with typical ones after the additive manufacturing of stainless steel. Moreover, the relationship between the structure relative density and its energy absorption capacity has been defined. The value of the maximum deformation energy depends on the adopted gradient topology and reaches the highest value for a gradually decreased topology, which also indicates the highest relative density. However, the highest rate of densification was observed for a gradually increasing topology. In addition, the results show that the gradient topology of the lattice structure affects the global deformation under the loading. Both, static and dynamic loading resulted in both barrel- and waisted-shaped deformation for lattices with an increasing and a decreasing gradient, respectively. Lattice specimens with a gradually changed topology indicate specific mechanical properties, which make them attractive in terms of energy absorption applications.

**Keywords:** lattice structures; additive manufacturing; selective laser melting; energy absorption; dynamic compression; split Hopkinson pressure bar

## 1. Introduction

Currently, development in the field of the additive manufacturing technique causes a growing interest in new multifunctional cellular materials [1–3]. Owing to the "layer by layer" method of fabrication and a wide spectrum of available materials (polymers, metals and resins) used in the manufacturing process, cellular materials with a regular structure indicate specific features unavailable to reach in comparison to bulk materials [3,4]. Low mass and high stiffness make them attractive as engineering materials [5]. They have been started to be used in this field of application in

leading branches of industry (automotive [6,7], aviation [8], railway [9], bio-engineering [10,11] and military [12–14]) as well as in civil engineering and modern art. Furthermore, high porosity resulted from a low value of relative density causes that they can be used as an effective thermal-, sound- and vibro-insulators [15,16]. Based on the data provided by many scientific papers, cellular materials are a very prospective group of engineering materials, which can go exceed the limit of typical bulk materials in the development of new cutting-edge products [17–19].

One of the key issues referring to the investigations of regular cellular material is the definition of the relationship between structure topology and its mechanical response in terms of energy absorption [20–23]. This issue attracts the attention of many research groups and has become an object of extensive studies carried out with the use of experimental [24,25] and numerical approaches [26–28]. Their results show that both cell topology and mechanical properties of used material strongly influence the mechanical behavior of cellular structures [28–30]. Taking into account modern technological possibilities of using AM methods, 2D and 3D variants of cellular materials can be produced. One of the well-known representatives of the 2D specimen group is honeycomb and its modifications. These structures indicate a high stiffness and they can be made with the use of all available 3D printing techniques from a wide spectrum of materials. The other group of cellular materials is represented by 3D lattice structures. They demonstrate significantly lower values of the relative density maintaining high mechanical strength, however, their fabrication process is more complex in comparison to honeycomb based structures and it demands more sophisticated AM techniques (selective laser melting—SLM, selective laser sintering—SLS, stereolithography—SLA, digital light processing—DLP and PolyJet). The main advantage of lattice structures is a great potential to make lightweight objects, optimization of mechanical constructions owing to the design freedom. Currently, progress in the field of metallic additive manufacturing systems, equipped in the powder bed feeding mechanisms (SLM, Direct Metal Laser Sintering-DMLS and Electron Beam Melting-EBM) and gradually decreasing costs of manufacturing process causes that many scientists focus their attention on the 3D cellular materials.

Xiao at el. [31] carried out investigations on a lattice structure with a rhombic dodecahedron topology made via EBM from a Ti-6Al-4V titanium alloy. They conducted experimental tests to determine the mechanical strength of material specimens under quasi-static loading at different temperatures. They compared the obtained results with the data gathered for stochastic metallic foams and found a similar mechanical behavior. Their results showed that the lattice structures are recommended as cellular materials dedicated to energy absorption applications. Mahshid et al. [32] evaluated the possibility of applying lattice structures as a solution to increase efficiency in tooling applications. Sign et al. [33] focused their attention on the manufacturability of lattice structures made from a titanium-tantalum alloy. They analyzed the mechanical properties of the manufactured specimens and their dimensional accuracy depending on the adopted SLM technological process parameters. They found that laser power is the most essential process parameter determining the final properties of the fabricated lattice structure. Based on the results presented in [34], it can be stated that one of the key issues related to the definition of lattice structure topology is sensitivity of a unit cell on the loading direction. The authors conducted extensive studies analyzing the influence of popular unit cell topologies on the structure deformation process. They discussed the main mechanisms responsible for structure damage during compression tests. Based on their results, it can be concluded that body-centered cubic (BCC) and face centered cubic (FCC) topologies are not aligned to the loading direction. This feature is especially important in terms of energy absorption applications; however, it limits the design freedom of a unit cell shape. This problem can be solved using topologies with the geometrical features of the unit cell changed gradually. Choy et al. [35] conducted a study on the mechanical behavior of additively manufactured graded lattice structures. They observed that the plateau stress and the value of energy absorption were significantly higher in comparison to uniform structures. Nevertheless, mechanical properties of the adopted building material Ti-6Al-4V titanium alloy resulted in abrupt shear failure with diagonal cracking across the whole structure. They

suggested that based on a more sophisticated density profile of the structure, it could be possible to program the deformation profile and a rate of energy absorption dedicated to specific applications.

Based on the results presented in the above-mentioned papers it could be stated that a gradually changed topology of lattice specimens may allow for a controlled deformation process of specimens subjected to loading conditions. This feature can be important in the terms of energy absorption in which the range of specimen capacity depends on the applied topology and the original material used during the process of manufacturing the specimens. Due to this reason, the authors decided to evaluate an influence of gradually changed variants of lattice specimens on their deformation history plots under both quasi-static and dynamic loading conditions. Selective laser melting has been chosen for this research as it involves a great variety of factors determining the final properties of components. These factors influence the density, microstructure and, hence, the mechanical properties of the final elements. Furthermore, the main advantage of the SLM is the ability to manufacture complex geometries, including 3D lattice structures. Taking into account the abovementioned possibilities of applying a selective laser melting technique and state-of-the-art in terms of additively manufactured lattice structures, the authors decided to conduct studies related to mechanical response of lattice structures with gradually changed topologies under both quasi-static and dynamic loading conditions. Based on the results presented in papers [34,36,37], the authors chose a standard body centre cubic (BCC) structure topology and, based on it, they developed gradient variants of structures presented in Figure 1.

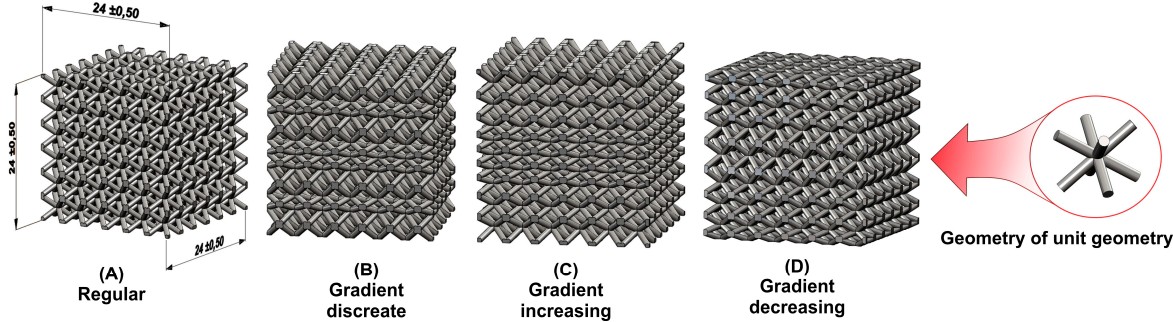

**Figure 1.** The general view of elaborated lattice structures.

The choice of the BCC elementary unit cell was justified with a specific behavior of the structure deformation process, which is dominated by bending [34]. This feature, in connection with high ductile metal materials, enables obtaining a long-range plateau crucial in terms of energy absorption, development of new protection systems dedicated to the military and civilian applications. Furthermore, additional variants of lattice specimens with gradually changed topologies were elaborated. All the models presented in Figure 2 were designed, with the same global dimensions (24 mm × 24 mm × 24 mm) and the same strut diameter (0.8 mm), in the SolidWorks CAD system. They differ in orientation of the gradually changed unit cells defined according to the method presented in Figure 2. In Table 1, a value of relative density for all examined cases is presented. The adopted global dimensions of specimens resulted from limitations of the laboratory set-up that was planned to be used to conduct compression tests under impact loading conditions.

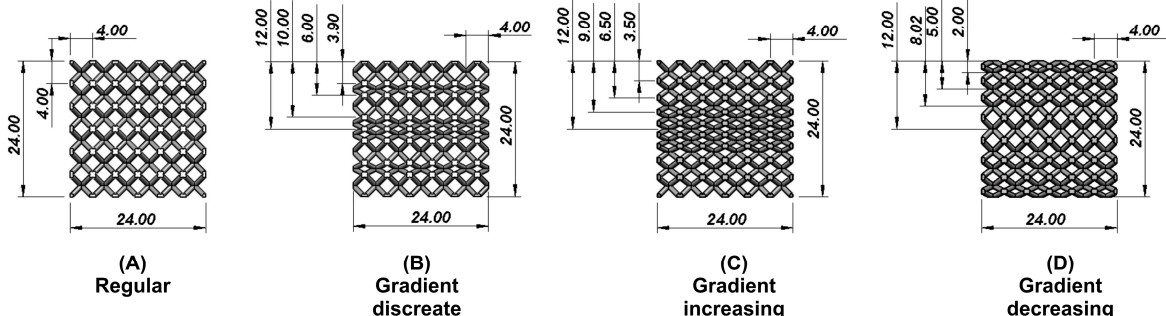

**Figure 2.** The detailed views of geometrical properties of a standard and a gradually changed lattice structure topologies.

**Table 1.** Main features of designed variants of lattice specimens.

| Variant of Lattice Specimen | (A) Regular | (B) Discrete Gradient | (C) Increasing Gradient | (D) Decreasing Gradient |
|---|---|---|---|---|
| Relative density (–) | 0.24 | 0.28 | 0.28 | 0.30 |

## 2. Lattice Specimens Manufacturing Process, Quality Control and Microstructure Studies

The gas atomized spherical SS316L powder (supplied by Material Technology Innovations C., Ltd., MTI S01) with the particle size in the range of 15–45 μm ($D_{10}$ = 17.24, $D_{50}$ = 28.92, $D_{90}$ = 44.32) has been used for manufacturing lattice structures. The chemical composition of the starting material in wt.% was ≤0.05% C, ≤0.03% S, ≤0.0045% P, ≤0.03% C, ≤1.0% Si, ≤2.0% Mn, 2.0–3.0% Mo, 10–14% Ni, 16–18% Cr and bal.% Fe. This powder, with flowability of <19 s/50 g, is characterized by excellent properties for additive manufacturing.

Selective laser melting was performed in an argon atmosphere with a pressure of 1000–2000 mbar using an AFS-M260 type system from the Beijing Company. The process parameters are pointed out in Table 2. These parameters were selected based on the literature analysis and are supposed to be optimal to obtain the highest density and low concentration of surface and microstructure defects. SLM, emerged in the late 1980s and 1990s, is known to be one of the additive manufacturing techniques. During the SLM process, the product is manufactured layer-by-layer through selective melting of the successive powder part using a laser. During the interaction between the laser and the powder, the powder material is heated until it melts and forms a liquid pool followed by the solidification and fast cooling down. After completing the scanning of the whole layer, the building plate is lowered down by one-layer thickness, a new layer of powder is deposited, and the process is automatically repeated until the part is built. It should be noted that before SLM process lattice structures with a gradient topology were designed using SolidWorks software (SolidWorks 2018, Dassalut Systems, Velizy-Villacoublay, France) followed by slicing the 3D CAD file data into the layers creating the stl file containing a 2D image of each layer using Materialise Magics 21.0 software (Materialise INC., Leuven, Belgium). Next, the stl file was loaded into a file preparation software package that assigns parameters, values and physical supports allowing the file to be interpreted and built with the SLM device.

**Table 2.** Selective laser melting (SLM) process parameters used for stainless steel 316L.

| | |
|---|---|
| Laser power (W) | 150 |
| Scan speed (mm/s) | 1000 |
| Beam diameter (μm) | 50 |
| Layer thickness (μm) | 30 |

Macroscopic observations are realized on a digital microscope (Figure 3) with an ability for visual inspection, failure analysis and quality control on raw (non-polished) samples. The other conducted

lattice structure quality control analyses associated with the evaluation of structural imperfections such as pores, voids and microcracks were performed using a Nikon Metrology XTH 225 CT. Microstructure evaluation is carried out using Phenom ProX/CeB$_6$ scanning electron microscope (SEM) (Thermo Fisher Scientific Inc., Eindhoven, The Netherlands) with an acceleration voltage at 15 kV equipped with an energy dispersive spectroscopy (EDS) chemical composition analyzer (Thermo Fisher Scientific Inc., Eindhoven, The Netherlands). Before microscopic observations obtained structures were sectioned, mounted in an epoxy resin and metallographically polished using silicon carbide abrasive paper and a diamond suspension (6, 3 and 1 μm) (Struers GmbH, Cracow, Poland). The final polishing is made using a silica suspension of 0.1 μm. To reveal the microstructure, the samples were etched with the use of Kalling's No. 2 reagent (CuCl$_2$ + HCl + C$_2$H$_5$OH). Electron dispersive X-ray (EDX) chemical composition mapping is performed for C, Fe, Cr, Ni, Mn and Mo elements. The elemental mapping of C was possible owing to the use of a silicon drift detector (SDD) (Thermo Fisher Scientific Inc., Eindhoven, The Netherlands) with a high sensitivity for light elements.

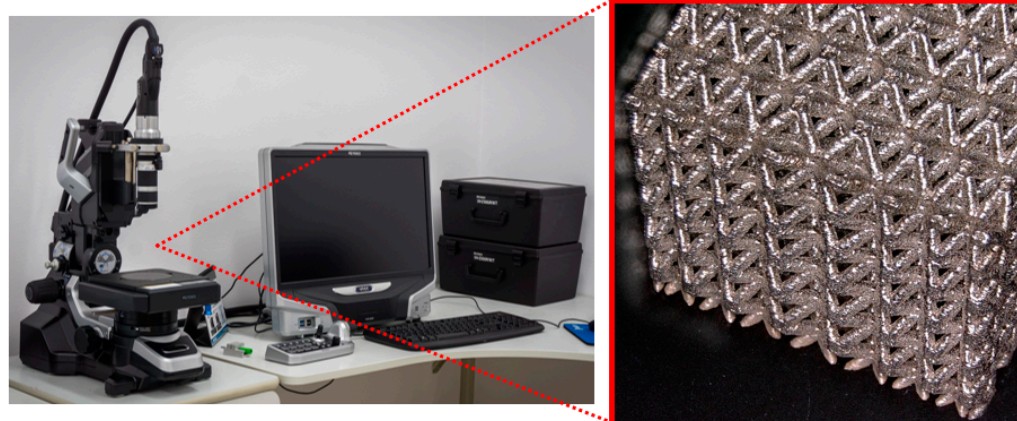

**Figure 3.** Digital microscope Keyence VHX-6000 (Keyence International, Osaka, Japan) used in the quality control process of manufactured structure specimens.

To validate the mechanical properties of the obtained SS316L structures, Vickers micro-hardness distribution measurements (Vickers Zwick/Roell) were conducted in both perpendicular and parallel directions to the layers with a load of 100 g and loading time of 10 s for every single indentation.

*2.1. Geometrical Quality Control of the Lattice Structures with Different Topologies*

To validate the dimensional mismatch between the designed and SLM manufactured lattice structures, additional quality control investigations were conducted. The struts thickness of all the proposed variants of the structure was measured with the use of the digital microscope presented in Figure 3.

In Figure 4, the frontal and side views of the regular lattice structure, as well as the front view of the structure with a discrete gradient together with the measured values of a strut diameter are shown. Referring to the results of measurements presented in Figure 4a,b, a clear difference of the geometrical deviations, depending on the measurement direction and the type of the specimen was observed.

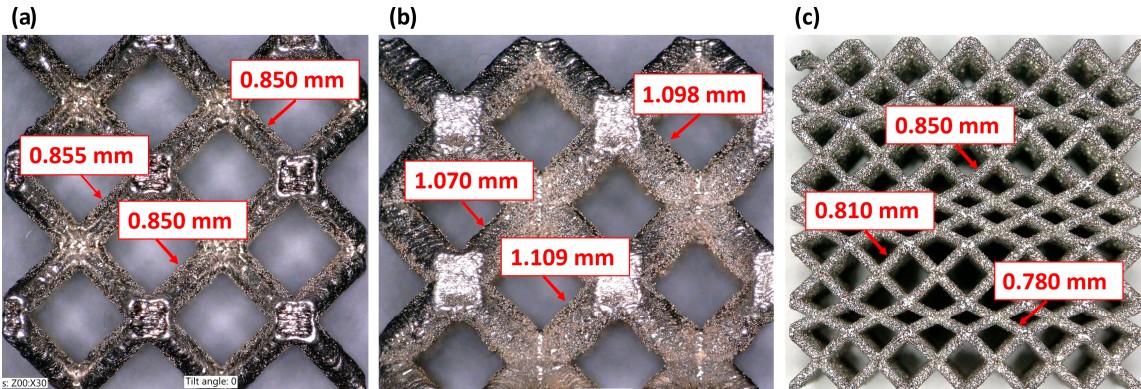

**Figure 4.** The dimensional deviations of strut thickness: (**a**) frontal-view of regular structure, (**b**) side-view of regular structure and (**c**) frontal-view of structure with the discrete topology.

The average dimensional deviation was about 0.05 mm and was influenced by the character of an additive manufacturing process and a grain size of the used SS316L powder. A detailed analysis demonstrated that the dimensional deviation was dependent on the side orientation—the dimensional deviations obtained from the side-view orientation were higher than those obtained from the frontal-view orientation. The dimensional deviations determined for a unit cell of a regular lattice structure were lower than 50 μm. As it can be seen in Figure 4c, the dimensional deviation of strut thickness was almost the same as for a regular structure obtained from the frontal view. The dimensional mismatches between the designed and the manufactured struts were generally caused by the adopted thickness of a melted layer of powder, and also resulted from laser power fluctuations during the SLM process. It should be noted that a similar variation between designed and as-built strut thickness was observed by Dallago et al. [38]. Referring to [38], the amount of excess material is dependent on the strut orientation to the printing direction.

In Figures 5 and 6, strut-level defects, generated during the SLM manufacturing process, which may affect the mechanical properties of the produced lattice structures, were depicted [39–41]. These strut-level defects are a variation in cross-sections along the strut length and the 'waviness' of the strut. The layer thickness of 30 μm, selected during the SLM process, influenced the surface roughness (variations in the strut's cross-section) among the struts. 'Waviness' is understood as the deviation of the strut axis across its length [40]. Both variation in cross-sections and 'waviness' contribute to the formation of local heterogeneities and stress concentrations that may affect the stiffness and compressive strength [40,42]. Furthermore, in Figure 5 it is clearly observed that a great number of adhered un-melted powder particles are bonded onto the struts of the lattice structures. This phenomenon is also observed by other researchers [43,44]. These particles are related to the partial melting of the raw powder particles on the boundary of the solid struts due to the contour laser track that was scanned only once [43]. Moreover, a significant difference in height was visible in the connection places of struts. These inequalities arise as a result of overlapping laser paths at the highest points, see Figure 6b. Such geometrical defects in the conjunction of the struts are likely to promote stress and strain localizations [45].

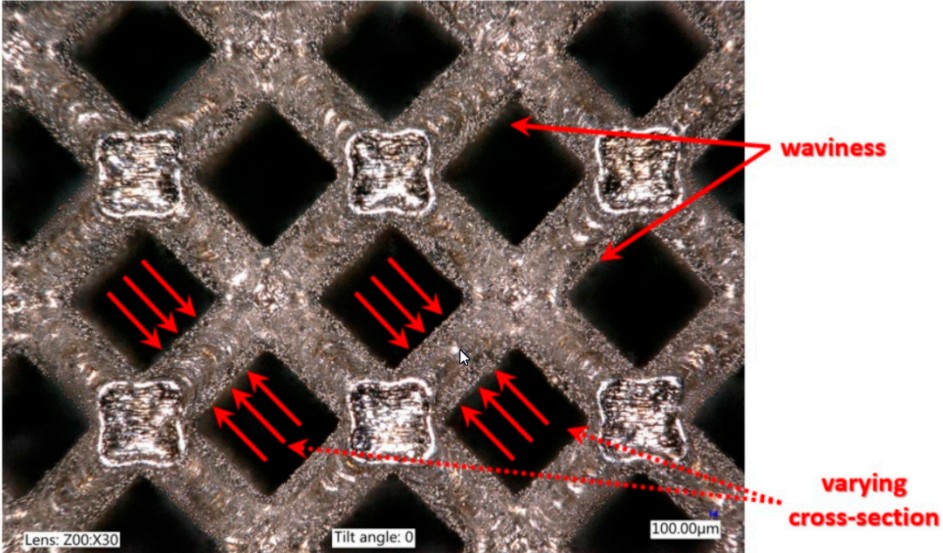

**Figure 5.** Strut-level defects generated during the SLM process.

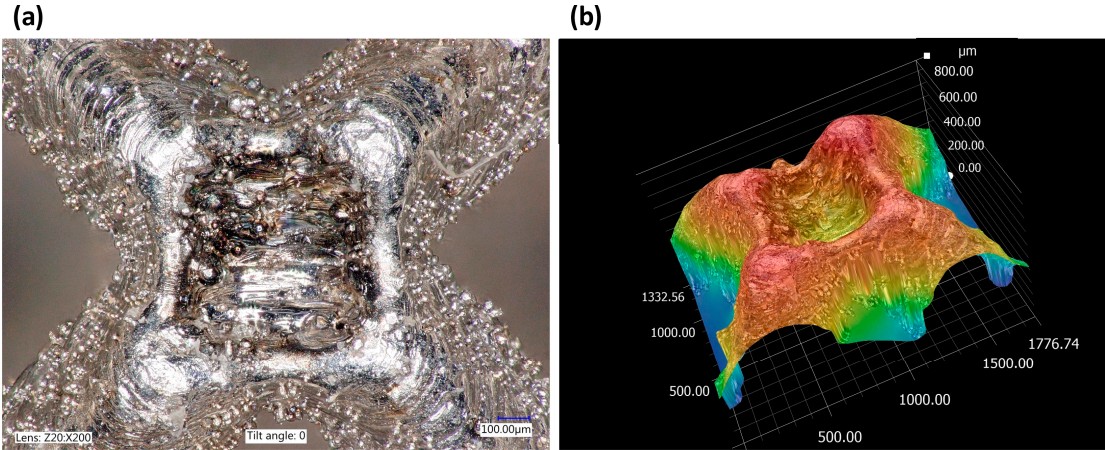

**Figure 6.** Optical micrographs showing the difference in height in connection places of struts: (**a**) general view of the connection place of struts, (**b**) differences in height in the connection place of struts.

### 2.2. Analysis Based on Computed Tomography

Computed tomography analysis demonstrated the occurrence of the structural imperfections in laser-melted specimens, such as porosity, voids and interconnectivity also observed in the authors' previous studies [25,46] and by other research groups [39,42,47,48]. The observed voids and pores were insignificant and located in various regions over the lattice, as it could be observed in Figure 7. Such imperfections are not dependent on the structure geometry but on the adopted manufacturing method. Considering the minor porosity and the high ductility of stainless steel, the authors found that the effect of these imperfections was negligible.

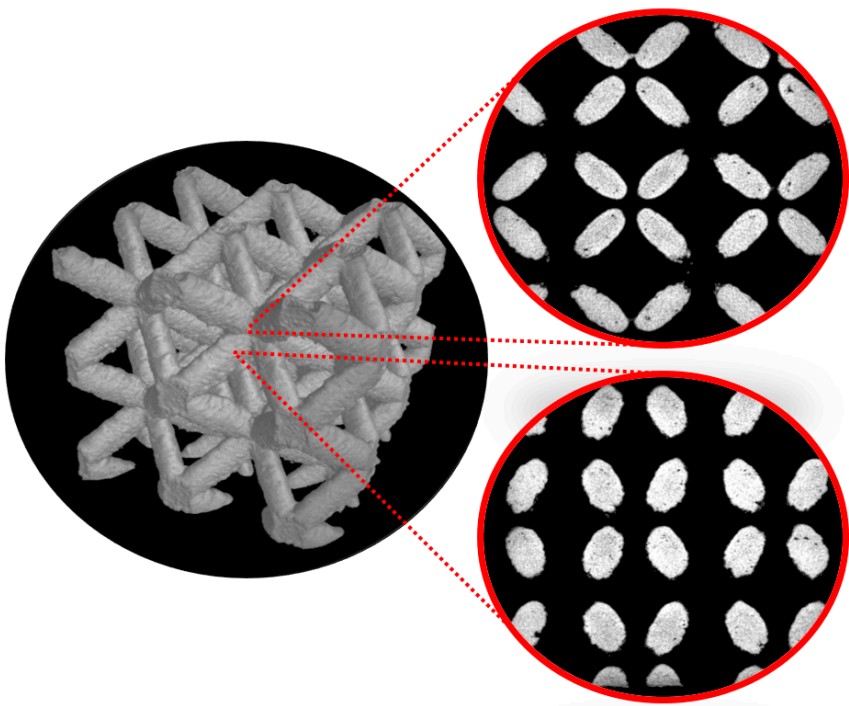

**Figure 7.** Computed tomography (CT) images of the SLM lattice structure.

## *2.3. Microstructure and Phase Composition*

The microstructure of the 316L stainless steel lattice structures manufactured using the SLM technique was described in detail in the authors' earlier work. Some defects such as pores, incomplete fusion holes and cracks were observed in produced specimens. Detected porosity appeared mostly in the form of pores small in size and spherical in shape, as it was shown, porosity in the produced structures was extremely low (below 0.5%), which results from insufficient powder packing during the SLM feeding process, so-called "inter-run porosity" or non-overlapped laser beam tracks. Lattice structures exhibited a hierarchical macro-, micro- and nanostructures (see, Figure 8) arising from non-equilibrium processes during SLM manufacturing and were composed of an array of elongated sub-grains inside coarse grains. Moreover, EDS analysis demonstrated a slight inhomogeneous distribution of Fe, Cr, Ni and Mo inside the grains. It was observed that also sub-grain boundaries are enriched in Mo. This inhomogeneous distribution is related to different solidification rates, even in the one melt pool, and to chemical composition fluctuations caused by the slow kinetics of large atoms [49,50]. The XRD analysis reveals the presence of a single face-centered cubic (fcc) austenite phase with a certain broadening of the peaks resulted from both lattice distortion induced by the SLM process and the presence of texture caused by highly directional growth [51].

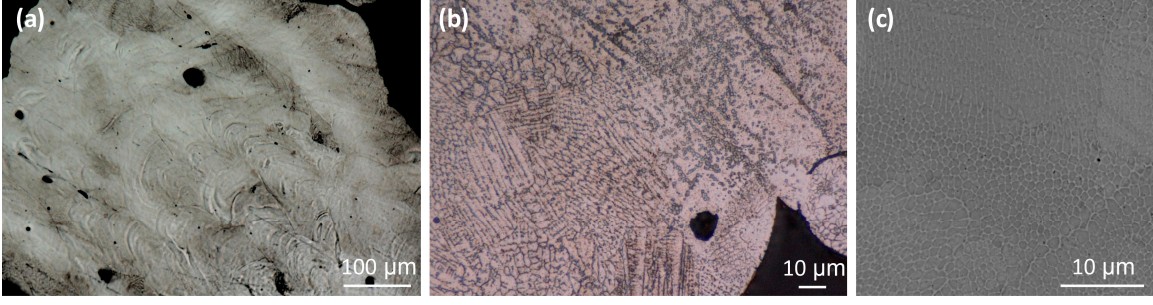

**Figure 8.** Optical (**a**,**b**) and SEM (**c**) images of the microstructure of SLM structures revealing the characteristic morphology of laser-melted techniques, i.e., equiaxed austenitic grains.

### 3. Investigation of Mechanical Behavior of Gradually Changed Lattice Structure

The manufactured lattice structure specimens described in the previous paragraphs were subjected to compression tests under quasi-static and impact loading conditions. The main aim of the conducted studies was the evaluation of the influence of a gradually changed topology on the deformation process. A compression test under quasi-static loading conditions enable a detailed observation of the main mechanisms responsible for the structure deformation, whereas, compression test under dynamic loading conditions allow the evaluation of inertia effects on the structure behavior under impact loading conditions.

### 3.1. Compression Tests under Quasi-Static Loading Conditions

A standard universal tensile machine MTS Criterion C45 (Figure 9) was used to perform tests under quasi-static loading conditions. The specimens were compressed at 1 mm/s velocity. The history of the deformation process was recorded by TW-Elite software (MTS System Corporation, Eden Prairie, MN, USA). Owing to the gathered data, a detailed analysis of the obtained results was possible. Analyzing the deformation force history plots presented in Figure 10, three main stages described in the literature [52,53] can be observed. The first one is a linear elastic deformation, followed by a plateau. After plateau, an increase of the loading force caused by the densification is observed. The first stage generally depends on the geometrical stiffness of the structure and mechanical properties of the applied 316L stainless steel material. The second stage is related to reaching the material yielding and presence of damage mechanisms such as bending and stretching. These mechanisms define the plateau deformation range. The final stage, in turn, refers to densification in which the structure is totally deformed, and original material is additionally compressed.

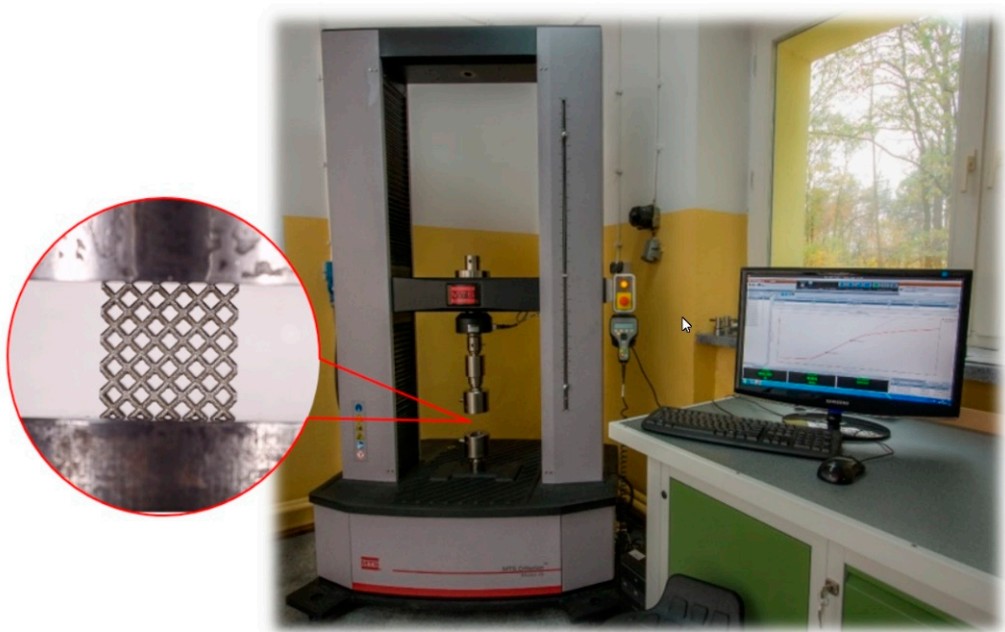

**Figure 9.** The general view of the universal tensile machine used to perform compression tests under quasi-static loading conditions.

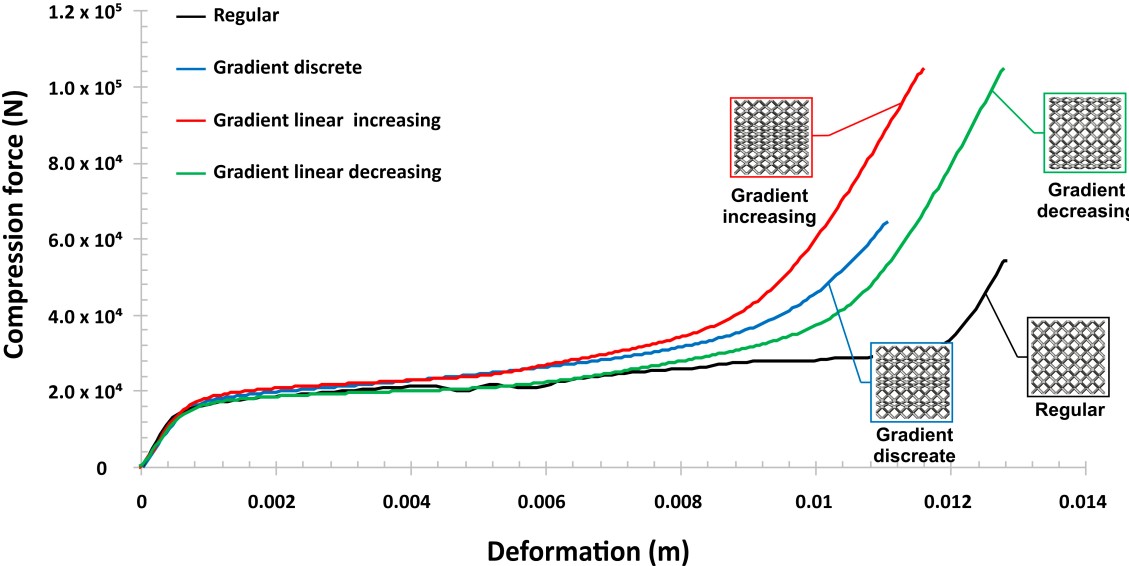

**Figure 10.** Deformation force versus displacement history plots registered for various lattice structure specimens.

Comparing the results presented in Figure 10, a considerable influence of a gradually changed topology on the specimen deformation process could be observed. Gradually changed topologies indicate a higher value of the relative density with regard to the regular one. This feature resulted in a lower range of plateau deformation and caused a significant increase in the deformation force. The highest range of the deformation force was noticed in the case of specimens with gradually increased and discrete topologies, which were characterized by a similar value of the relative density. A value of the maximum deformation force registered in the case of a decreased gradient was similar as in the case of a regular topology. Depending on the orientation of the applied gradients (increase, decrease and discrete), it could be observed that a gradually increased topology caused a progressive increase in the deformation force during the plateau range, which results in the highest value of both the deformation force (Figure 9) and the deformation energy (Figure 11).

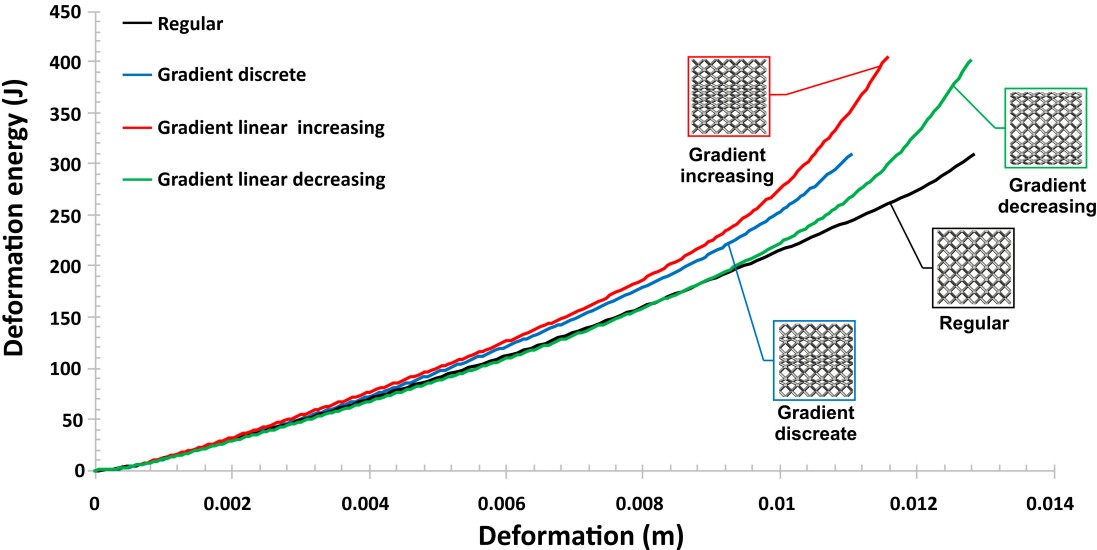

**Figure 11.** Deformation energy versus displacement history plots registered for various lattice structure specimens.

Additionally, in Figures 12–14, the main stages of regular and gradient topologies (linearly increasing and decreasing) during the deformation process were presented. Analyzing the photographs presented below, it can be observed that specimens with a regular topology during compression deformed uniformly. Specimens with gradually changed sizes of the elementary unit cell, in turn, deformed depending on the gradient orientation. A specimen with an increasing gradient (Figure 13) took the concave form during compression, whereas a specimen with a decreasing gradient took an oblique shape (Figure 14).

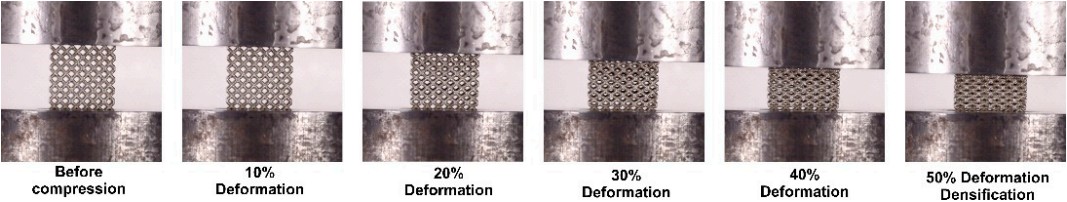

**Figure 12.** The main stages of the deformation process of a lattice specimen with a regular topology.

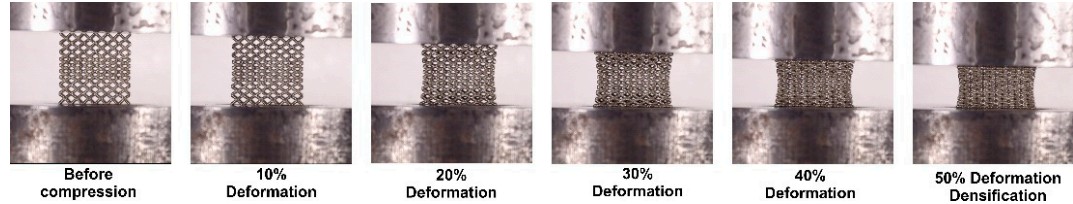

**Figure 13.** The main stages of the deformation process of a lattice specimen with a gradually increasing topology.

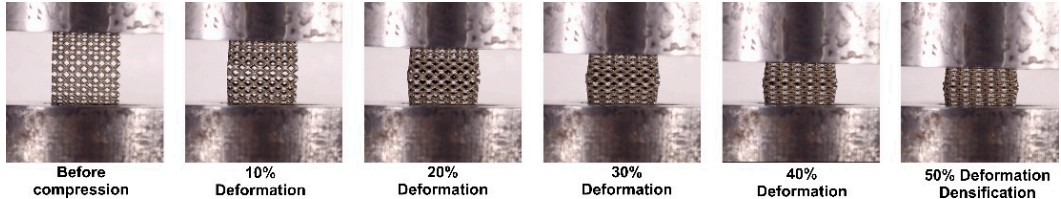

**Figure 14.** The main stages of the deformation process of a lattice specimen with a gradually decreasing topology.

### 3.2. Compression Tests under Impact Loading Conditions

In the next stage of the investigation, the mechanical behavior of lattice specimens manufactured additively from 316L stainless steel was defined based on the compression tests under impact loading conditions. A split Hopkinson pressure bar laboratory set-up presented in (Figure 15) was used to perform these studies. The principle of operation of the SHPB is well known and it was discussed in numerous works [54–56]. The SHPB apparatus is generally used to perform the identification of mechanical properties of materials under a high-strain rate. However, after the application of bars with a larger diameter and adequately longer length, this set-up enables conducting studies concerning the mechanical response of cellular material under impact loading conditions. Additionally, compared to the drop tests, it allows increasing the range of strain rate from $10^2$ to $10^3$ s$^{-1}$. The SHPB apparatus with a 40 mm diameter, presented in Figure 15, was set up to conduct tests in a direct impact scenario, in which the specimen was located on the impact surface of the input bar and subjected to the direct impact of the striker. This configuration of the SHPB apparatus resulted in higher strain rate effects and enabled an increase in the range of specimen plastic deformation. The tests were performed with an initial velocity of striker changing from 10 to 12 m/s. Depending on the applied specimen topology, a value of the impact velocity necessary to cause full densification of the structure was defined.

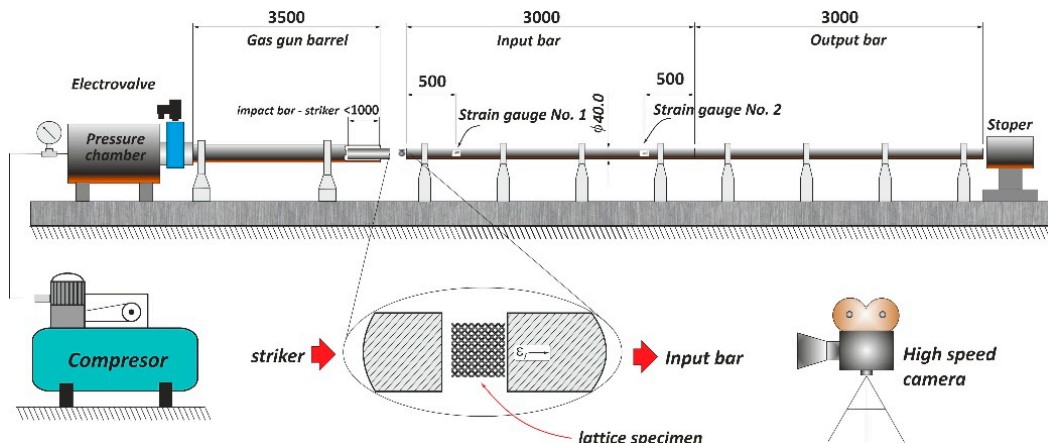

**Figure 15.** The general view of the split Hopkinson pressure bar laboratory set up used to carry out impact loading tests.

Based on the conducted SHPB tests, the results presented in Figures 16 and 17 were obtained. Analyzing the deformation force curves (Figure 16), similar plots registered for all variants can be observed. The deformation process consists of three main stages, which were described in the previous paragraph. Referring to the deformation force versus a displacement curve, there was a visible similarity in linear elastic and plateau ranges of the deformation process. However, after the beginning of the densification a difference between gradually increasing and decreasing structures could be observed. In the case of a gradually increasing specimen, this process began quicker in comparison to the other variants, which was caused due to a considerable number of layers with a small size of elementary unit cells with the highest geometrical stiffness. The specimen with a gradually decreasing topology was characterized by the highest deformation rate, which resulted in the maximum value of deformation energy (Figure 17). Furthermore, this specimen, which was characterized by the highest value of the relative density, had a significant effect in the case of dynamic tests.

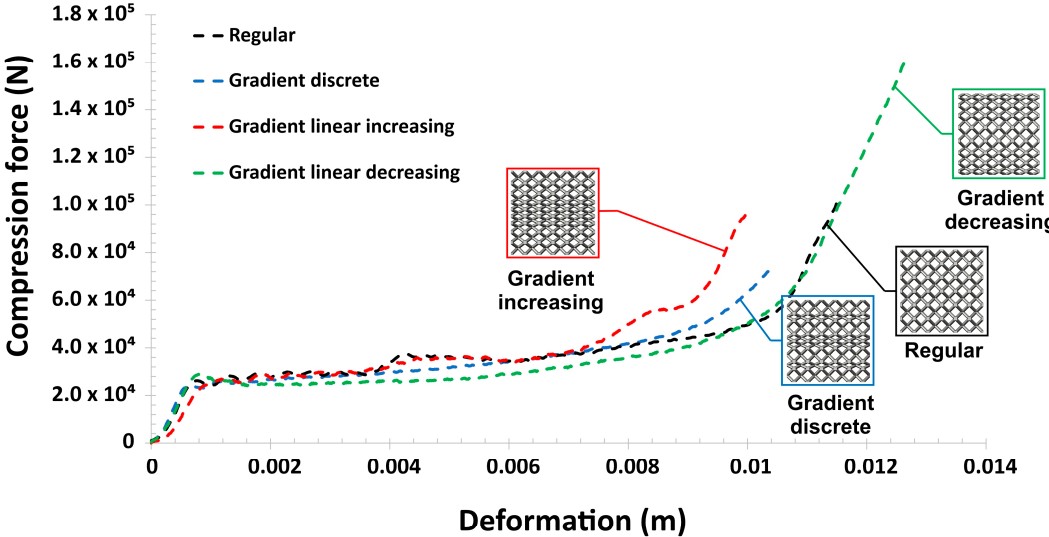

**Figure 16.** Results of compression tests (deformation force versus displacement) conducted under impact loading conditions.

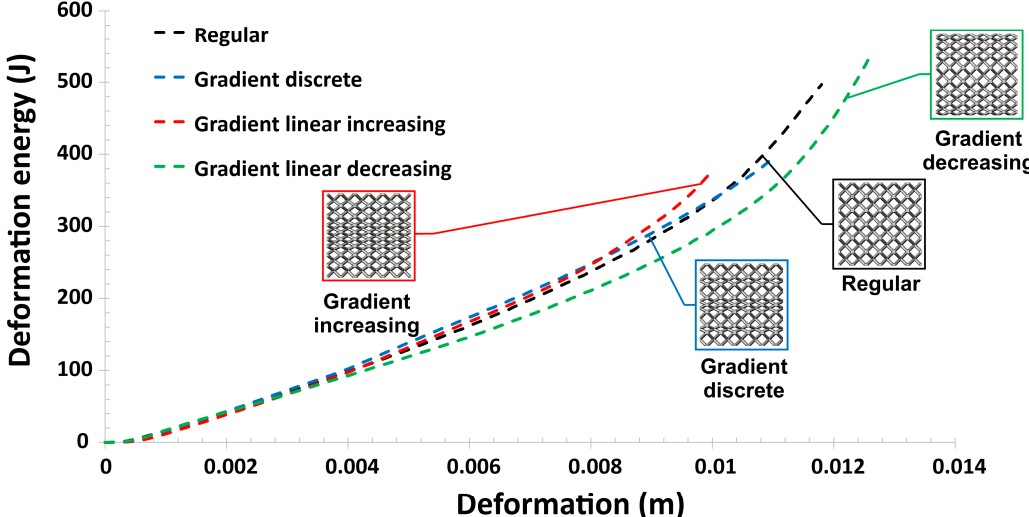

**Figure 17.** Results of compression tests (deformation energy versus displacement) conducted under impact loading conditions.

## 4. Discussion

Figure 18 presents the comparison of deformation of all lattice structures (regular and gradient ones) after both static and impact compression tests. These results show an evident difference in the global deformation of gradient structures. The regular structure deformed in a typical linear uniform way—the sidewalls of these structures are parallel to each other. The structure with a discrete gradient demonstrated deformation almost the same to the above-mentioned, whereas the gradient lattice structures behaved differently. For a regular structure, linear deformation happens if all unit cells of lattice structure are symmetrical (or slightly asymmetrical). In the structure with an increasing gradient, when loaded under the compression, the lateral deformation was outward in the top part, and the bottom part of the structure it was much higher than for the middle part of the structure. In turn, the gradient decreasing lattice structure under compression exhibited local bulking in the middle part of the structure, which results from higher outward deformation at this place. These two lattice structures showed narrow-waisted and barrel-shaped deformations. A similar relationship was noticed by Hedayati et al. [57].

It is worth noting that the Vickers hardness of all the lattice structures increased from 238 to 285 HV for the as-built state and after Hopkinson testing, respectively. The increase in the Vickers hardness corresponded well to the structure evolution and to an increase in the number of dislocations that arose during the dynamic loading. Additionally, in Figures 19 and 20, the comparison of the deformation energy results obtained under quasi-static and dynamic tests was presented. Based on the obtained data, a visible influence of strain rate effects could be observed. The values of the deformation energy registered during the SHPB tests were higher referring to the results gathered under quasi-static loading conditions, excluding the variant with a gradually increasing topology. Taking into consideration the value of deformation, it could be found that, during quasi-static compression tests, a rate of densification was controlled and the tests were carried out until high peak of deformation force was obtained. Analyzing an influence of a gradually changed topology on the structure deformation process, a possibility for programming this process depending on an adopted topology could be observed (Figure 21). This issue was significantly important in terms of the energy absorption application as well as crashworthiness.

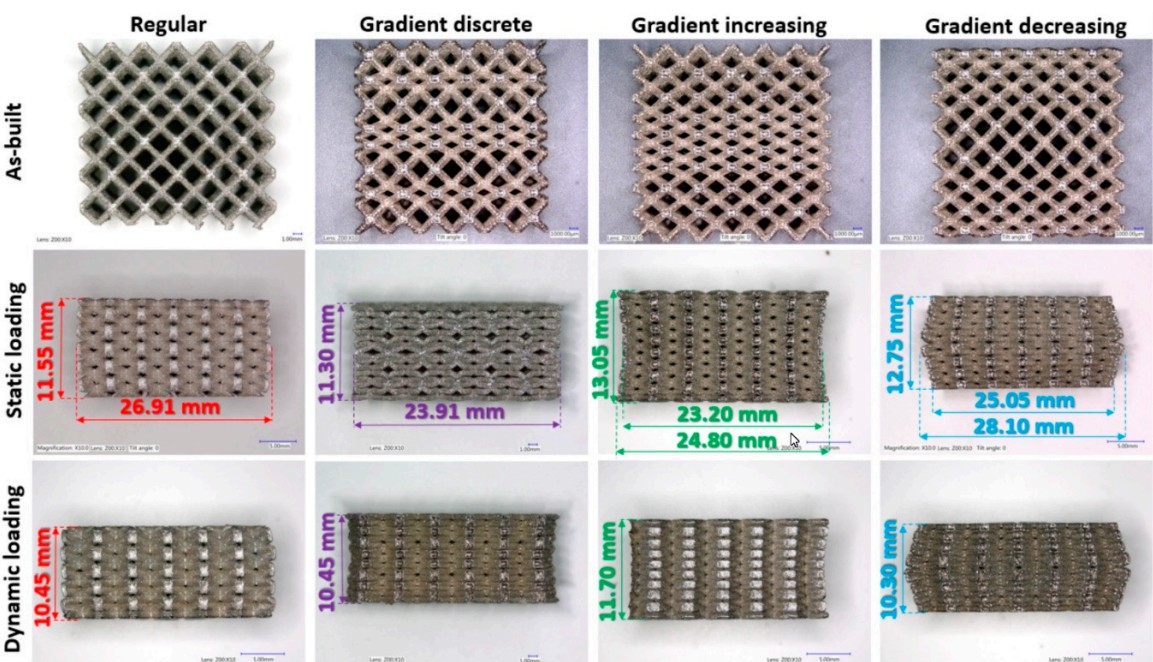

**Figure 18.** The deformation of lattice structures with different topologies after static and dynamic testing.

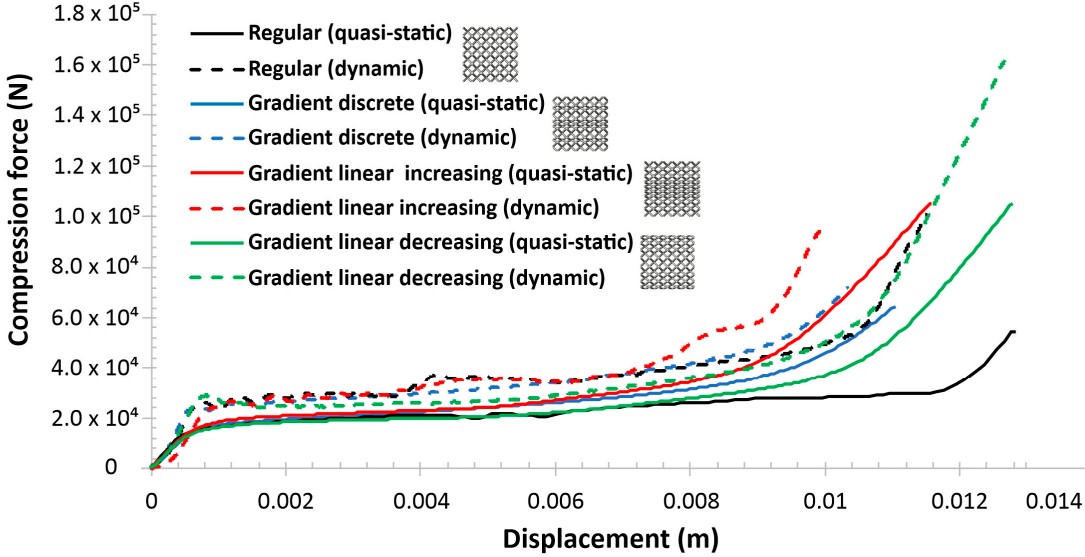

**Figure 19.** Comparison of compression force history plots obtained in quasi-static and impact compression tests.

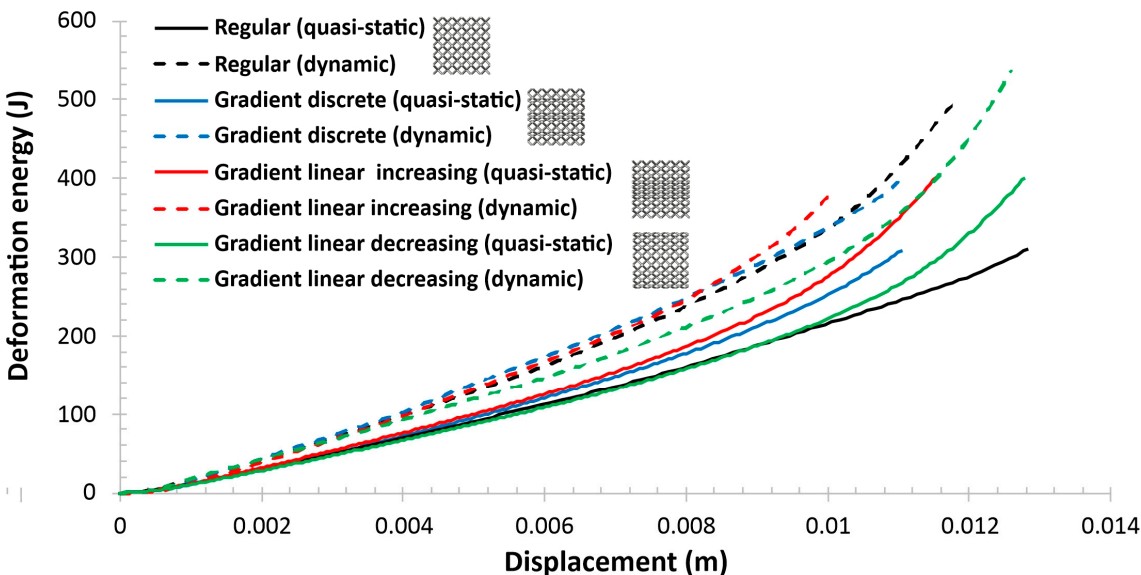

**Figure 20.** Comparison of deformation energy history plots obtained in quasi-static and impact compression tests.

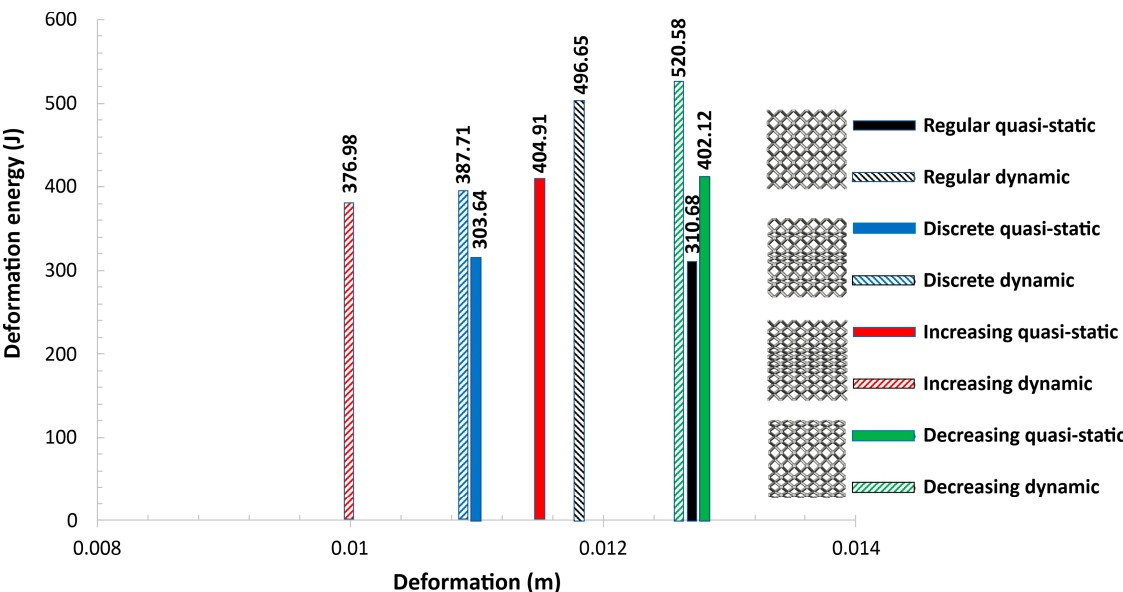

**Figure 21.** Comparison of the deformation energy values obtained in quasi-static and impact compression tests.

## 5. Conclusions

In the presented study, lattice structures with regular and gradient topologies (discrete, decreasing and increasing) were designed and fabricated additively from 316L stainless-steel powder using the SLM technique. The manufactured lattice structure specimens were evaluated in terms of quality control and were subjected to both static and impact compression tests to define their mechanical behavior and main differences in the deformation process. Based on the conducted works, the following conclusions can be drawn:

I.    The adopted additive technique SLM (selective laser melting), with the technological parameters defined based on the literature, enabled the fabrication of the designed lattice structure specimens made from the 316L stainless steel powder. However, evaluations of the geometrical accuracies and microstructures revealed some weaknesses of the selected manufacturing process, such as

the deviation of the dimensions of lattice struts measured on the frontal plane (perpendicular to material deposition layers) and on the side planes as well as a geometrical deviation, e.g., waviness and varying cross-section along the strut.

II. The evaluation of the material structure with the use of CT (computed tomography) revealed the presence of imperfections such as porosity, voids and the presence of unmelted powder grains. However, the authors found that the effect of these imperfections was negligible under static and dynamic compression tests.

III. The SLM-produced structures exhibited a typical microstructure for additive manufacturing. The revealed microstructure was composed of only an austenitic phase. The precise microstructure exhibited the presence of hierarchical macro-, micro- and nanostructures that arose during the SLM process. Based on the XRD pattern, it was confirmed that austenite is a dominant phase.

IV. Compression tests performed under quasi-static loading conditions showed that specimens with a gradually changed topology indicate a different behavior during the deformation process. Depending on the adopted topology, various values of the maximum deformation energy could be obtained. Furthermore, the application of a gradually increasing topology enabled obtaining a higher rate of densification. The deformation process was similar as in the case of auxetic structures.

V. The results of compression tests under impact loading conditions revealed the strain rate sensitivity of the lattice specimens. The maximum value of the deformation energy was registered in the case of the specimen with a gradually decreased topology, which indicates the highest value of the relative density. Furthermore, the behavior of specimens during the deformation process was similar as in the case of quasi-static investigations.

VI. Investigations need to be continued to improve the technological process parameters to enable the improvement of the geometrical quality of lattice specimens as well as to reduce structural and microstructural imperfections. In addition, numerical studies were planned to be performed to define the mechanical responses of lattice structure specimens with a wide range of geometrical parameters.

VII. Gradient lattice structures made of SS316L demonstrated behavior that made them attractive as a prospective, new light engineering material with a high mechanical strength. High ductility of the applied SS316L stainless steel in combination with a cellular structure could be very effective in terms of energy absorption applications. Further development would enable their application to further cutting-edge products in many industrial fields.

Additively manufactured 3D lattice structures exhibited different deformation behavior dependent on a gradient topology under both static and dynamic loading. It was found that the global deformation of the structure was closely associated with the applied gradient variant. However, future work is required to include other parameters, e.g., different cell structures or gradient variants. It seems to be worth to evaluate the mechanical response of specimens in which the geometrical properties of strut are the main parameter that determines the gradual topologies. These kind of studies are presented in papers [45,58]. Furthermore, in the future, the results presented in the paper was planned to be extended on numerical studies, including optimization procedures, which would enable finding the most effective topology of the lattice structure.

**Author Contributions:** J.S. participation in lattice specimen manufacturing process, conducting the microstructure and microscopy analysis, preparation of the final manuscript. P.P. design of lattice specimens, conceptualization of the investigations, analysis results of quasi-static and dynamic tests, participation in preparing the final manuscript. A.R. analysis of the data and approval of the final version of the manuscript. F.J. responsible for the specimen manufacturing process, optimization of the SLM technological process parameters. X.S. manufacturing the samples. All authors have read and approved the content of the manuscript.

**Funding:** The research was funded by the Polish Ministry of Science and Higher Education under the research grant no. PBS/23-883/2019/WAT.

**Acknowledgments:** The authors would like to thank the following persons for contributing to performing the compression tests: Jacek Janiszewski, Piotr Dziewit, Marcin Sarzynski, and Kamil Cieplak. Additionally, we would like to thank our colleagues from the Institute of Machinery Equipment Design, Mechanical Engineering Department, Military University of Technology, for their suggestions and remarks regarding the manufacturing process.

**Conflicts of Interest:** The authors declare no conflict of interest.

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
