# Peer review of "Investigations on the Mechanical Response of Gradient Lattice Structures Manufactured via SLM"

_metals, doi:10.3390/met10020213_

Round 1
Reviewer 1 Report
Additive manufacturing of gradient lattices has been under investigation from last few years and this research work has been done in the same area of additively manufactured gradient lattice structures. Moreover, a great effort has been made to investigate mechanical behavior of the structures. But some minor issues have been observed after carefully observing the manuscript which are listed below. After careful modifications, this manuscript can be considered for possible publication. Please find suggestions and comments below.
Title of the manuscript seems little confusing for potential reader. Please simplify it. Possible alternative titles could be, e.g., “Investigations on the mechanical response of gradient latticestructures additively manufactured via SLM”, “Investigations on the mechanical response of additively manufactured gradient lattice structures” or “Investigations on the mechanical response of gradient lattice structures manufactured via SLM”. Please describe in brief, “why SLM has been chosen over other technologies for this particular research work”. First sentence of abstract is grammatically not correct. Please modify it. There are some error messages in manuscript, please correct it (L133). The basic design followed by the authors is not clear, whether they have taken it from somewhere or they have followed some other research work. Though it has been written that authors have used CAD package to design it. Please clear this and mention the basic internal structure of the design. In line 185, authors have cited wrong figure, it should be figure 5 c. In line 278, please correct (figure 13-15). In section 2.3, authors have found some porosity due to uneven rate. Which is expected in this process. So, what measures did authors take to overcome this issue? Conclusion must be summarized, and future considerations must be briefly discussed after conclusion. A discussion about the extension of the presented results to the design of graded, additively manufactured pentamode structures made of metallic materials would significantly enrich the applicative potential of the presented research (refer, e.g., to Experimental response of additively manufactured metallic pentamode materials confined between stiffening plates. Composite Structures, 142, 254–262, 2016. Mechanical modeling of innovative metamaterials alternating pentamode lattices and confinement plates. Journal of the Mechanics and Physics of Solids, 99, 259-271, 2017).
Author Response
Dear Reviewer,
Thank you for your useful comments and suggestions on our manuscript.
The manuscript has been modified accordingly, and detailed corrections are listed below point by point:
1. Title of the manuscript seems little confusing for potential reader. Please simplify it. Possible alternative titles could be, e.g., “Investigations on the mechanical response of gradient lattice structures additively manufactured via SLM”, “Investigations on the mechanical response of additively manufactured gradient lattice structures” or “Investigations on the mechanical response of gradient lattice structures manufactured via SLM”.
Title of the manuscript has been changed to “Investigations on the mechanical response of gradient lattice structures manufactured via SLM”.
2. Please describe in brief, “why SLM has been chosen over other technologies for this particular research work”.
It has been described why SLM has been chosen for this research.
3. First sentence of abstract is grammatically not correct. Please modify it.
The sentence has been corrected.
4. There are some error messages in manuscript, please correct it (L133).
The abovementioned sentence has been corrected.
5. The basic design followed by the authors is not clear, whether they have taken it from somewhere or they have followed some other research work. Though it has been written that authors have used CAD package to design it. Please clear this and mention the basic internal structure of the design.
Presented in the following paper results of investigations are a continuation of authors previous works which were subjected to publishing and currently are still under review. The BCC Body-Center-Cubic unit cell topology were proposed due to its simplicity. Authors previous works contained results were different unit cell size and diameter of strut was evaluated. Based on obtained results a 4 mm height and 0.8 strut diameter parameters were selected as a referential topology and based on it, gradient topologies were designed.
6. In line 185, authors have cited wrong figure, it should be figure 5 c.
It has been corrected.
7. In line 278, please correct (figure 13-15).
It has been corrected.
8. In section 2.3, authors have found some porosity due to uneven rate. Which is expected in this process. So, what measures did authors take to overcome this issue?
Detected porosity, mostly in form of small of size and spherical in shape, can be connected with:
- insufficient powder packing during SLM,
- so-called “the inter-run porosity” - spaces between the melt pools between successive laser passes,
- the areas of not overlapped laser beam tracks.
Observed porosity is very low – below 0.5 % - and cannot be completely eliminated.
9. Conclusion must be summarized, and future considerations must be briefly discussed after conclusion.
It has been added.
10. A discussion about the extension of the presented results to the design of graded, additively manufactured pentamode structures made of metallic materials would significantly enrich the applicative potential of the presented research (refer, e.g., to Experimental response of additively manufactured metallic pentamode materials confined between stiffening plates. Composite Structures, 142, 254–262, 2016. Mechanical modeling of innovative metamaterials alternating pentamode lattices and confinement plates. Journal of the Mechanics and Physics of Solids, 99, 259-271, 2017).
Authors are thankful for the recommendation of mentioned works. They are very interesting and demonstrate additional potential in terms of definition specimens with gradual topology, where the main parameters subjected to study will be geometrical properties of the strut. Referring to technological possibilities of the SLM technique, mentioned variants of structures can be elaborated and subjected to compression tests.
At least, we are very thankful for your time and efforts in handling our manuscript.
Judyta Sienkiewicz

Reviewer 2 Report
There are many grammatical errors and inappropriate words in the paper. For example, in the first sentence of the abstract, “the main aim of the paper is evaluation the mechanical behavior or lattice specimens subjected to quasi-static compression tests.”should be changed to “the main aim of the paper is to evaluate the mechanical behavior of lattice specimens subjected to quasi-static compression tests.” In the third paragraph of the Introduction, the authors only listed the previous investigations, without any summary. Besides, the significance of the study was not mentioned in the Introduction. The captions of Figure 2B and Figure 2D are wrong (page 3). Figure 2B should be Gradient decreasing, and Figure 2D should be Gradient discrete. In Section 2.3 (page 8), is there any effect of the inhomogeneous distribution of the chemical compositions? The caption of Figure 9 is wrong (page 9), “ Optical (a, b) and SEM (b) images of the microstructure of SLM structures revealing the characteristic morphology of laser-melted techniques, i.e. equiaxed austenitic grains “ should be “ Optical (a, b) and SEM (c) images of the microstructure of SLM structures revealing the characteristic morphology of laser-melted techniques, i.e. equiaxed austenitic grains “ In the second paragraph of Section 3.1 (page 9), the statement “ The highest range of deformation force was noticed in the case of specimens with gradually increased and decreased topologies which are characterized by a similar value of the relative density. The value of maximum deformation force registered in the case of a discrete gradient was similar like in the case of regular topology. “ is wrong. According to Figure 11, it should be, “The highest range of deformation force was noticed in the case of specimens with gradually increased and discrete topologies which are characterized by a similar value of the relative density. The value of maximum deformation force registered in the case of a decreased gradient was similar like in the case of regular topology. ” In Figure 15 (page 11), please present the deformation history of the specimen with the discrete gradient core. Though the final deformation shape can be found in Figure 19, it is expected to add the photographs of the deformation history here. In Figure 17 (page 13), what is the reason for the fluctuations in the force-deformation curves of the specimen with the gradient increasing core? Figure 21 (page 15) is redundant and can be deleted. On the one hand, the comparison of deformation energy at different displacements is meaningless. On the other hand, the comparison is shown in Figure 20. It will be better to add specific energy absorption figures for both quasi-static and dynamic tests. The description of the deformation and force-deformation curves should be more comprehensive. For example, for quasi-static tests, the forces in the densification stages varied significantly, and the densification stages occurred at different displacements in the four specimens. These were not mentioned in the paper. Since deformation/displacement is small, please use “mm” instead of “m” for figures such as Figs. 18 and 20.
Author Response
Dear Reviewer,
Thank you for your useful comments and suggestions on our manuscript.
The manuscript has been modified accordingly, and detailed corrections are listed below point by point:
1. There are many grammatical errors and inappropriate words in the paper. For example, in the first sentence of the abstract, “the main aim of the paper is evaluation the mechanical behavior or lattice specimens subjected to quasi-static compression tests.” should be changed to “the main aim of the paper is to evaluate the mechanical behavior of lattice specimens subjected to quasi-static compression tests.”
The sentence has been corrected.
2. In the third paragraph of the Introduction, the authors only listed the previous investigations, without any summary. Besides, the significance of the study was not mentioned in the Introduction.
Thank you for your suggestion. Appropriate modification of the text was introduced to conclude the introduction part and point out the main aim of this work.
3. The captions of Figure 2B and Figure 2D are wrong (page 3). Figure 2B should be Gradient decreasing, and Figure 2D should be Gradient discrete.
Thank you for your comment. The captions were changed in modified version of the paper.
4. In Section 2.3 (page 8), is there any effect of the inhomogeneous distribution of the chemical compositions?
The observed inhomogeneity is small enough to not affect the deformation process during both static and impact loading.
5. The caption of Figure 9 is wrong (page 9), “ Optical (a, b) and SEM (b) images of the microstructure of SLM structures revealing the characteristic morphology of laser-melted techniques, i.e. equiaxed austenitic grains “ should be “ Optical (a, b) and SEM (c) images of the microstructure of SLM structures revealing the characteristic morphology of laser-melted techniques, i.e. equiaxed austenitic grains “
It has been corrected.
6. In the second paragraph of Section 3.1 (page 9), the statement “ The highest range of deformation force was noticed in the case of specimens with gradually increased and decreased topologies which are characterized by a similar value of the relative density. The value of maximum deformation force registered in the case of a discrete gradient was similar like in the case of regular topology. “ is wrong. According to Figure 11, it should be, “The highest range of deformation force was noticed in the case of specimens with gradually increased and discrete topologies which are characterized by a similar value of the relative density. The value of maximum deformation force registered in the case of a decreased gradient was similar like in the case of regular topology. ”
It has been corrected.
7. In Figure 15 (page 11), please present the deformation history of the specimen with the discrete gradient core. Though the final deformation shape can be found in Figure 19, it is expected to add the photographs of the deformation history here.
The reviewer suggestion is appropriate. Unfortunately, during quasi-static compression tests of specimens with a discrete gradient the process of deformation was not registered due to the mistake of memory card. It was planned to fill lack figures in, however, in a short period of time it is impossible. Authors analyzed the recorded film of the discrete specimens deformation process under dynamic loading conditions and found that this process was similar to the variant with regular topology. For this reason, the decision was undertaken that figures presenting a detailed view of the deformation process will be limited to regular, gradually increasing and gradually decreasing topologies.
8. In Figure 17 (page 13), what is the reason for the fluctuations in the force-deformation curves of the specimen with the gradient increasing core?
Thank you for your remark. In the author's opinion, these fluctuations were caused by the progressive deformation of the specimen's struts and subsequent deformation of building material after the partial densification of the specimen.
9. Figure 21 (page 15) is redundant and can be deleted. On the one hand, the comparison of deformation energy at different displacements is meaningless. On the other hand, the comparison is shown in Figure 20. It will be better to add specific energy absorption figures for both quasi-static and dynamic tests. The description of the deformation and force-deformation curves should be more comprehensive. For example, for quasi-static tests, the forces in the densification stages varied significantly, and the densification stages occurred at different displacements in the four specimens. These were not mentioned in the paper. Since deformation/displacement is small, please use “mm” instead of “m” for figures such as Figs. 18 and 20.
Referring to the Reviewer remarks authors decided to divide the Figure 20 on two separate figures. This decision was justified by the fact that based on presented plots it is a visible difference between data resulted from the strain rate effect which arrive under impact loading conditions. This effect is very important in term of energy absorption and in the author's opinion, it is worth to present it. Regarding the issue of units (recommended mm milimeter), authors used "m - meter" in all charts to avoid misunderstanding in definition the value of deformation energy, Force (N) x Displacement (m) = Energy (J).
At least, we are very thankful for your time and efforts in handling our manuscript.
Judyta Sienkiewicz
Reviewer 3 Report
The paper is acceptable from the scientific point of view, but it needs an extensive editing of the English language and style. The authors are suggested to ask the help of a native English speaker for correcting the manuscript (there are too many language errors to be mentioned in this review).
Please also take note of the following errors:
the affiliation of the third author should be "3", not "4" the captions of Figures 1 and 2 should be accompanied by bibliographic references Line 133 - on page 4 contains a broken bibliographic reference some values included in Figures 5 and 19 are hardly readable the arrows included in Figure 6 are not visible the text of the legends in Figure 20 should be written using a larger font to increase the readability reference 38 (Line 518 on page 19) should be accompanied by a description of the website content.
Author Response
Dear Reviewer,
Thank you for your useful comments and suggestions on our manuscript.
The manuscript has been modified accordingly, and detailed corrections are listed below point by point:
1. The affiliation of the third author should be "3", not "4"
It has been corrected.
2. the captions of Figures 1 and 2 should be accompanied by bibliographic references
Figures 1 and 2 were made by authors.
3. Line 133 - on page 4 contains a broken bibliographic reference
Could you please indicate exact which bibliographic reference is broken.
4. some values included in Figures 5 and 19 are hardly readable
It has been corrected.
5. the arrows included in Figure 6 are not visible
It has been corrected.
6. the text of the legends in Figure 20 should be written using a larger font to increase the readability 
Mentioned Figure 20 was divided into two separate figures to better present the strain rate effect under dynamic loading conditions.
7. reference 38 (Line 518 on page 19) should be accompanied by a description of the website content.  
It has been corrected.
At least, we are very thankful for your time and efforts in handling our manuscript.
Judyta Sienkiewicz
Reviewer 4 Report
It is an interesting paper.
I think that waht is missing a clear comparison of your defects with the one's described in the papers by Prof. Pasini (but perhaps his density is much lower) and the recent ones by Boniotti.
It would be interesting to see where your data would appear in Ashby's maps in terms of Young's modulus and yield strength.
It would be also interesting to have more info about the damage mechanism: failure occur in the struts or the nodes ?
Author Response
Dear Reviewer,
Thank you for your useful comments and suggestions on our manuscript.
The manuscript has been modified accordingly, and detailed corrections are listed below point by point:
1. I think that waht is missing a clear comparison of your defects with the one's described in the papers by Prof. Pasini (but perhaps his density is much lower) and the recent ones by Boniotti.
It has been added.
2. It would be interesting to see where your data would appear in Ashby's maps in terms of Young's modulus and yield strength.
The mechanical properties of applied 316L stainless steel as a building materials were as follow: Young Modulus – 190 GPa , Yield strength – 540 MPa. Taking into consideration the relative value of the density it was between 0.28 to 0.3 [-].
3. It would be also interesting to have more info about the damage mechanism: failure occur in the struts or the nodes ?
Based on a detailed evaluation of deformed specimens it was found that damage mechanism was generally initiated in nodes. The geometrical stiffness of strut cause that the bucking and bending mechanisms arrived at the beginning. A subsequent stage of the deformation caused occurring of the stretching mechanism in nodes and afterwards in struts.
At least, we are very thankful for your time and efforts in handling our manuscript.
Judyta Sienkiewicz
Round 2
Reviewer 2 Report
The authors have amended the manuscript according to the reviewers' comments.
Author Response
Dear Reviewer,
Thank you very much for your comments.
Yours sincerely,
Judyta Sienkiewicz
Reviewer 3 Report
Definitely, the new version of the article has been significantly improved by the authors and I would like to congratulate them for the hard work.
There is only one error remaining in the text regarding the broken reference which was not solved in the new version of the article:
see line 149 (page 4) where it is mentioned: "parameters are pointed out in Error! Reference source not found.2."
"Error! Reference source not found.2." should be replaced with the with the corespondent bibliographic source which is missing!
After this minor change, article can be accepted for publication in this form!
Author Response
Dear Reviewer,
Thank you very much for your comments.
The error you have noticed has been corrected.
Yours sincerely,
Judyta Sienkiewicz